# Mapping the CP-Transgene Insert in the Papaya Genome and Developing a Hermaphrodite Transgenic Hybrid with Broad-Spectrum Resistance to Papaya Ringspot Virus

**DOI:** 10.3390/v16060823

**Published:** 2024-05-22

**Authors:** Shyi-Dong Yeh, Ya-Chi Lin, Ching-Shan Tseng, Chih-Chi Liao, Chung-Hao Huang, Shin-Lan Wang, Ya-Ling Huang, Chia-Peng Chang

**Affiliations:** 1Department of Plant Pathology, National Chung Hsing University, Taichung 402, Taiwan; isvdic10809@gmail.com (Y.-C.L.); chichiphysics@gmail.com (C.-C.L.); anthelloween@hotmail.com (C.-H.H.); nchuk02087@nchu.edu.tw (S.-L.W.); tina9050@yahoo.com.tw (Y.-L.H.); apon48970119@yahoo.com.tw (C.-P.C.); 2Advanced Plant and Food Crop Biotechnology Center, National Chung Hsing University, Taichung 402, Taiwan; 3Crop Genetic Resources and Biotechnology Division, Taiwan Agricultural Research Institute, Taichung 413, Taiwan; samtseng@tari.gov.tw

**Keywords:** papaya ringspot virus, transgenic resistance, marker-assisted breeding, transgenic papaya hybrid

## Abstract

Papaya ringspot virus (PRSV) limits papaya production worldwide. Previously, we generated transgenic lines of hybrid Tainung No.2 (TN-2) carrying the coat protein (CP) gene of PRSV with broad resistance to PRSV strains. Unfortunately, all of them were female, unacceptable for growers and consumers in practical applications. With our reported flanking sequences and the newly released papaya genomic information, the CP-transgene insert was identified at a non-coding region in chromosome 3 of the papaya genome, and the flanking sequences were verified and extended. The female transgenic line 16-0-1 was first used for backcrossing with the parental Sunrise cultivar six times and then followed by selfing three times. With multi-level molecular markers developed from the PRSV CP transgene and the genomic flanking sequences, the presence and zygosity of the CP transgene were characterized at the seedling stage. Meanwhile, hermaphrodite genotype was identified by a sex-linked marker. With homozygotic transgene and horticultural properties of Sunrise, a selected hermaphrodite individual was propagated by tissue culture (TC) and used as maternal progenitor to cross with non-transgenic parental cultivar Thailand to generate a new hybrid cultivar TN-2 with a hemizygotic CP-transgene. Three selected hermaphrodite individuals of transgenic TN were micropropagated by TC, and they showed broad-spectrum resistance to different PRSV strains from Taiwan, Hawaii, Thailand, and Mexico under greenhouse conditions. The selected clone TN-2 #1, with excellent horticultural traits, also showed complete resistance to PRSV under field conditions. These selected TC clones of hermaphrodite transgenic TN-2 provide a novel cultivation system in Taiwan and elsewhere.

## 1. Introduction

Papaya (*Carcica papaya* L.) is an important fruit crop with high economic value in the tropics and subtropics. However, commercial papaya production is limited by the disease caused by papaya ringspot virus (PRSV) worldwide [1,2]. The virus was first recorded in major plantation areas of southern Taiwan in 1975 [3]. Since then, it has spread throughout Taiwan, and currently, papaya plants must be planted under netting to exclude aphid vectors to ensure production [1,4].

Transgenic resistance for the practical control of plant virus diseases has been developed for several crops, such as squash [5,6], cantaloupe [7], melon [8], and watermelon [9]. The transgenic papaya lines carrying the coat protein (CP) of PRSV [10,11] were first generated by transformation through particle bombardment [12]. The SunUp and Rainbow varieties of papaya have been commercialized as the first case of transgenic fruit crops to ease the threat of PRSV in Hawaii [13,14,15]. However, these transgenic cultivars only confer resistance to the PRSV Hawaii strain, not other geographic strains [1,4]; thus, their deployment in other geographic regions is hampered.

*Agrobacterium*-mediated transformation based on wounding zygotic embryos in the liquid phase was applied to transform the papaya hybrid cultivar Tainung No.2 (TN-2) with the CP gene of the Taiwan strain PRSV YK for the generation of the first transgenic papaya in Taiwan [16]. Results from tests under greenhouse conditions indicated that transgenic papaya line 16-0-1 provides complete resistance to the severe strain of Taiwan and broad-spectrum resistance to other geographic strains from Hawaii, Mexico, and Thailand [17].

Unfortunately, transgenic papaya line 16-0-1 and other CP-transgenic lines developed in Taiwan are all females [17,18,19] and not practically useful because they need pollination from male flowers to set fruits. In contrast to pear-shaped hermaphrodite fruits, female fruits are round, less sweet, thin-fleshed, and have a large cavity with many seeds; therefore, consumers do not prefer them, and they are not acceptable to growers [20,21]. Incorporating the transgenic resistance of 16-0-1 [16,17,18] by breeding to generate a hermaphrodite papaya of Taiwan’s most popular hybrid cultivar TN-2 to meet the market and plantation demands becomes a priority.

The molecular markers specific for tracking the CP-transgene of transgenic papaya line 16-0-1 in the progenies of the crossing were previously established [19]. These molecular markers have two levels of specificity, including transgene-specific markers to trace the transgene [16,17] and event-specific markers using the genomic sequences flanking the transgene to identify and differentiate specific transgenic lines [19]. The draft genome of the transgenic papaya SunUp was first released in 2008 [22]. Recently, the chromosome-level genome (GenBank assembly accession GCA_022788785.1) of non-transgenic Sunset papaya has been reported, containing one sex chromosome (HSY), nine autosomes, and 57 unplaced contigs [23]. The new genomic information provides a valuable resource for comparative genomic analyses [24,25]. In this report, the previously obtained genomic sequences adjacent to the CP-transgene of the transgenic line 16-0-1 [19] were used to locate the transgene in the improved papaya genome, and longer genomic sequences flanking the transgene were identified.

Another function of the event-specific markers using the primers designed from the genomic sequences flanking the transgene is to monitor the hemizygote or homozygote status of the transgene in individual plants. Using primer pairs designed from the flanking sequences across the T-DNA insert, which contains the *npt* selection marker and the CP-transgene, no PCR product can be amplified from the homozygote progenies of line 16-0-1 because the large T-DNA is inserted at both alleles and the targeted T-DNA contexts are too long for generating amplification. In contrast, the presence of the specific PCR product can be detected from 16-0-1 hemizygotes in which one non-transgenic allele contains no T-DNA insert [19]. Hence, new primers were designed from the extended flanking sequences to help verify the status of homozygotes during the breeding process.

After a genetic linkage map was constructed [26], the molecular markers of the papaya genome were developed. Using random amplified polymorphic DNA (RAPD) analysis, 12 RAPD markers are associated exclusively with the hermaphrodite plants. Among those, the W11 marker is the most closely linked to the sex determination loci, and the specific primer pair of W11 (primers W11-F and W11-R) for identifying hermaphrodite plants at the seedling stage has been verified [27]. Thus, the W11 marker was used in this investigation to promptly select hermaphrodite papaya plants during backcrossing and selfing in the breeding process.

In this investigation, we first incorporated the PRSV CP transgene of the female 16-0-1 line into the Sunrise cultivar, one of the parental lines of TN-2, to fix the horticultural Sunrise properties and produce hermaphrodite individuals with the CP transgene for further crossing. After six times backcrossing to acquire the Sunrise criteria, the selected hermaphrodite plants were further self-crossed three times to generate a transgenic Sunrise with homozygotic CP-transgene, which was then used as a maternal line to cross with another parental cultivar, Thailand, to generate a new transgenic TN-2 hybrid.

During the breeding process, we applied PCR-based techniques using PRSV CP transgene-specific primers to identify the transgenic progenies. Meanwhile, the sex marker W11 was used to identify the hermaphrodite sex. Finally, the previously identified [19] and newly extended flanking sequences in this study were used to select homozygotic-transgene progenies during the selfings of Sunrise as a mother line for generating transgenic TN-2 hybrids. With the help of different molecular markers described above, we present the long-term breeding process, more than ten years, for developing a new transgenic papaya hybrid TN-2. Three selected hermaphrodite individuals of transgenic TN-2 with top fruit quality, namely TN-2 #1, 5, and 7, were mass-propagated by tissue culture (TC) [17,28]. Under greenhouse conditions, the three TC clones were tested against different geographic PRSV strains from Hawaii, Thailand, and Mexico. Moreover, TN-2 #1 plants were further evaluated under field conditions.

Our selected three hermaphrodite TC clones of the transgenic TN-2 hybrid confer complete resistance against different geographic strains of PRSV under greenhouse conditions. The TN #1 clone also provides complete protection against PRSV under field conditions in Taiwan. Here, we present a novel plantation model using TC clones of selected hermaphrodite individuals of the new transgenic papaya TN-2 with broad-spectrum resistance to PRSV strains to produce homogenous high-quality papaya fruits in Taiwan and elsewhere.

## 2. Material and Methods

### 2.1. Mapping of Flanking Sequences to the Papaya Genome

The 349.88 Mb sequence of the reference papaya genome (GCA_022788785.1) [23] was retrieved from the NCBI database. The flanking sequences of transgenic papaya 16-0-1 from our previous study [19] were used as queries for alignment to the genomic reference by command-line BLAST analysis. Another genome assembly (GCF_000150535.2) with available annotation was also used in BLAST to confirm the position of flanking sequences in the papaya genome.

The 500 upstream and downstream nucleotides extended from the previously identified short flanking sequences [19] were retrieved from the genomic sequences using the BEDTools software (version 2.30.0) package [29]. On the extended flanking sequences (30,828,249 to 30,828,748 and 30,829,353 to 30,829,852 base of chromosome 3) identified in this study, new primers (sequences of Papa31ex and Papa57ex as shown in Figure 1c) were used for double confirmation of the selecting of individuals containing hemizygous or homozygous PRSV CP transgene, following the previously described method [19]. The homozygotes with large inserts at both alleles do not exhibit PCR amplification by the flanking primers because the T-DNA insert is too long and blocks the amplification. In contrast, the hemizygotes with one allele without the T-DNA inserts can be amplified by the two flanking primer pairs.

### 2.2. The Breeding Process for Backcrossing, Selfing, and Hybridization

The test plants of transgenic line 16-0-1, a female TN-2 hybrid that contains a single PRSV CP transgene and confers broad-spectrum resistance to different geographic PRSV strains [16,17], were used for backcrossing with the TN-2 parental Sunrise cultivar. All plantlets, including those of transgenic 16-0-1, non-transgenic hermaphrodite or female parental Sunrise cultivar, and another parental line of TN-2 of Thailand cultivar, were multiplied by in vitro shoot-tip micropropagation as previously described [17,28]. According to governmental regulations, all seedlings were incubated in a temperature-controlled greenhouse (25 ± 3 °C) certified for genetically modified crops (GMO) and then transplanted at the 20 cm stage at the government-certified isolated field for transgenic crops at the Bei-Gou experimental farm of National Chung Hsing University (NCHU). Backcrossing and selfing were conducted by manual pollination. Seedlings of the seeds collected from fruits at each breeding step were incubated in the government-certified greenhouse for GMO at NCHU.

After six backcrosses and three times selfing, the hermaphrodite individual of Sunrise 2210 with homozygous CP-transgene was selected and used as the maternal line to cross with the non-transgenic parental Thailand cultivar to generate a new transgenic hybrid TN-2 with female or hermaphrodite sex. The inheritance of the CP-transgene, the hermaphrodite sex of individuals, and the zygosity status of the selected individuals at the seedling stage were identified by the molecular markers described below.

### 2.3. Precision Selection Assisted by Transgene-Specific, Event-Specific, and Hermaphrodite Markers

The forward primer pPYCP213 (5′-AGAGGCATACATCGCGAAGA-3′) and the reverse primer mPYCP213 (5′-CTGCCGTCCATTCCAAACAT-3′), reflecting the PRSV YK CP sequence [30], were used to amplify a 213 bp PCR product of the PRSV CP transgene to examine its existence after crossings.

Following the positive results of the transgene-specific detection, the primers Papa31 and Papa57 designed from the flanking sequences of the CP insert of the 16-0-1 line were used to identify the hemizygote (with 227-bps PCR product) or homozygote (without 227-bps PCR product) status of the CP transgene, according to the previous study [19]. In addition, new molecular markers (Papa31ex and Papa57ex to amplify 855-bps product from non-transgenic allele) designed from the extended flanking sequences in this study were also used to verify the homozygosity of the CP transgene.

The primer pair W11 (W11-F and W11-R) [27] was used to distinguish female (without 833-bps PCR product) or hermaphroditic (with 833-bps PCR product) individuals.

### 2.4. Micropropagation of the Selected Individuals by Tissue Culture

The original female transgenic TN-2 16-0-1, the finally selected female Surise 2210 with homozygous CP-transgene, the non-transgenic parental Thailand cultivar of TN-2, and the developed transgenic TN-2 (hermaphrodite) in this investigation were micro-propagated following the shoot-tip tissue culture (TC) method as previously described [28]. After rooting and acclimatization, the established plantlets of each clone were incubated in a temperature-controlled GMO greenhouse (25 ± 3 °C) for further tests.

### 2.5. DNA Purification and PCR Amplification

The genomic DNA from the papaya leaf (0.1 g) was isolated by DNeasy Plant Mini Kit (Qiagen, Valencia, CA, USA) according to the manufacturer’s instructions. The quality and quantity of each DNA sample were calibrated using a micro-volume spectrophotometer (NanoPhotometer N60, Implen, Munich, Germany) to ensure adequate input DNA for further experiments. Twenty nanograms of each DNA sample (with final concentration of 0.5-1.0 ng/μL) was used in a PCR reaction of total 20 microliter (μL) volume containing 0.2 μM specific primers, 150 μM dNTP, and 0.8 units of ProTaq™ DNA Polymerase (Protech, Taipei, Taiwan) in PCR buffer (1.5 mM MgCl_2_, 10 mM Tris-HCl, pH 9.0, 50 mM KCl, 0.01% (*w*/*v*) gelatin and 0.1% Triton X-100). PCR amplification was performed in a SimpliAmp thermal cycler (Applied Biosystems, Thermo Fisher Scientific, Waltham MA, USA) programmed with an initial denaturing step at 95 °C for 3 min, followed by a total of 30 cycles of denaturing at 95 °C for 30 s, annealing at 52 °C for 30 sec and extension at 72 °C for 1 min, and terminated with a 5-min extension step at 72 °C. The final PCR products were analyzed by electrophoresis on 1% agarose gel.

### 2.6. Resistance Evaluation of the Hermaphrodite Transgenic TN-2 Hybrid against Different Geographic PRSV Isolates

Three hermaphrodite individuals of the new transgenic TN-2 developed in this study were selected and propagated in TC, and their clones were designated as TN-2 #1, 5, and 7. To test the viral resistance of transgenic lines, individual papaya plants of Sunrise 2210 with homozygous CP-transgene, TN-2 #1 (Sunrise 2210 X Thailand) with hemizygous CP-transgene, and non-transgenic Tainung No.2 (TN2), at the six-to-eight leaf stage, each with 10 plants, were inoculated with different strains of PRSV from Taiwan (YK), Hawaii (HA), Mexico (MX), and Thailand (TH) [17]. The inocula were prepared from 0.1 g of papaya leaves of TN-2 infected with individual strains at 14 dpi (days post-inoculation) in 10 mL of 0.01 M potassium phosphate buffer (pH 7.2). The inoculated plants were kept in a temperature-controlled isolated GMO greenhouse (25 ± 3 °C, without light supplementation). At 28 dpi, nine leaf disks (0.5 cm in diameter) were punched from each tested plant’s three upper-fully-expanded leaves (three discs per leaf). The collected samples were assayed by indirect ELISA using the antiserum to PRSV CP [31] and by RT-PCR using 10-fold-diluted 1 μg cDNA as template for PRSV CP detection [16]. The experiments were independently repeated three times.

### 2.7. Field Test of the Hermaphrodite Transgenic TN-2 Hybrid

The field trials were conducted in the GM-crops-specific isolated fields of NCHU, certified by the Council of Agriculture of Taiwan. The field trial was conducted from 2020 to 2021 in a randomized complete block design (RCBD) in triplicate. Because papaya needs to be grown for about 7 months for harvesting fruits, a one-year period can observe the complete phase of papaya cultivation. Moreover, plantation of papaya in November avoided heavy rains and tropical storms in Taiwan. Each block included 20 transgenic and 20 non-transgenic plants, with a total of 60 TC-propagated transgenic TN-2 # 1 plants and 60 non-transgenic TN-2 control plants. All papaya plants were planted 2.5 and 2 m between and within rows, respectively. The test field was exposed to natural infection by aphid vectors; no artificially inoculated or diseased plants were provided in the field. The disease incidences of the trial were monitored by symptom development every month [18], and the occurrence of PRSV was checked using indirect ELISA, as described above.

## 3. Results

### 3.1. Identification of the Genomic Location of the CP-Transgene Insert of Line 16-0-1

The 236 and 353 bp genomic sequences flanking the CP transgene of line 16-0-1 were previously identified [19], but their exact positions in the papaya genome were not known. Here, we used the 353 bp sequence flanking the right border of T-DNA and the 236 bp sequence flanking the left border to align with the exact position in the papaya genome assembly (GCA_022788785.1) [22,23]. The result indicated that both are in chromosome 3 positioned approximately at 30.83 mega-base (Mb) (Figure 1a); the 353 bp is aligned from 30,828,749 to 30,829,101 bases and the 236 bp one is from 30,829,117 to 30,829,352 bases; a small deletion (15 bp) from 30,829,102 and 20,829,116 base was noticed.

Further checking the genomic annotation, the PRSV CP transgene of 16-0-1 was found inserted in the non-coding region of the intergenic region of the LOC110819880 and LOC110819887 genes, as illustrated in Figure 1b. LOC110819880 has been annotated to produce an uncharacterized protein, and LOC110819887 encodes the DEAD-box ATP-dependent RNA helicase 5 of the NW_019014676 contig (Figure 1a), which is assembled in chromosome 3 [22,23]. The 500 bases of extended genomic sequences upstream (30,828,249 to 30,828,748 bases) and downstream (30,829,353 to 30,829,852 bases) to the previously identified flanking sequences were retrieved from NCBI, and additional primers Papa31ex and Papa57ex (black arrows in Figure 1c) were designed to verify the transgene’s genomic location.

### 3.2. Creation of Hermaphroditic Sunrise with Homozygous Transgene as a Parental Line

The whole breeding process to generate the PRSV-resistant hermaphrodite TN-2 papaya is illustrated in Figure 2. The female 16-0-1 was backcrossed once with selected hermaphrodite plants of the Sunrise cultivar to produce BC1 hermaphrodite papaya plants carrying a hemizygous PRSV CP transgene. The selected BC1 hermaphrodite individuals with the transgene were used to further backcross with the non-transgenic Sunrise female five times. Hermaphrodite Sunrise individuals with hemizygous PRSV CP transgene and phenotypes similar to Sunrise were selected. The selected individuals of the transgenic Sunrise with hemizygous transgene were self-crossed once to generate Sunrise papaya carrying homozygous CP transgene. The selected Sunrise plants were further self-crossed twice to fix the horticultural properties of Sunrise, designated as line Sunrise 2210 in Figure 2.

### 3.3. Precision Breeding Assisted by Molecular Markers

The selection of the transgenic individuals, assessment of their zygosity status, and the confirmation of hermaphrodite sex were conducted using molecular markers. The transgene-specific (PRSV CP by pPYCP213 and mPYCP213 in Figure 3a) and event-specific primers of Papa31 and Papa57 [19] were used to identify the transgene and its hemizygous or homozygous status of progenies at the seedling stage of each breeding course. Following the identification of the presence of CP transgene (PCR product of 213 bp) from the transgenic individuals, the PCR products of 227 bp amplified by the two event-specific primer pairs designed from the flanking sequences indicated the presence of a non-transgenic allele, reflecting the hemizygous genotype (representative illustrations in Figure 3a; labeled as 16-0-1 in Figure 3b and S2210xT in Figure 3c). The two new flanking primers Papa31ex and Papa57ex designed from the extended flanking sequences, identified in this study (Figure 3a), produced an 857 bp product, and thus verified the hemizygote or homozygote status checked by the primer pair Papa31/Papa57. The negative amplification results with the two flanking primer pairs from CP-transgene positive individuals indicated the homozygote status of the transgene allele (labeled as S2210 in Figure 3b,c) because the distance between the two flanking primer pairs is too far to amplify a product across the T-DNA insert. In the absence of CP product, the positive amplifications with the two flanking primer pairs reflected the non-transgenic DNA samples (labeled as Non-transgenic in Figure 3).

With further use of W11 sex-linked primer pair (W11-F/W11-R) [27], which amplified a product of 833 bp, the hermaphroditic sex of the progenies of each backcrossing or selfing in the breeding process was also promptly identified at the seedling stage (Figure 2 and Figure 3).

To breed the transgenic TN-2, the female Sunrise 2210 with homozygous transgene was used as the maternal line to cross with another paternal line of hermaphroditic non-transgenic Thailand cultivar (Figure 2). The resulting TN-2 hybrid carrying a hemizygous PRSV CP transgene (the final thick-framed rectangle in Figure 2) was verified by PCR (Figure 3c).

### 3.4. Micro-Propagation of Selected TN-2 Hermaphroditic Individuals for Practical Application

Tissue cultured (TC) seedlings (Figure 4a–d) for various papaya varieties, including transgenic Sunrise 2210, transgenic TN-2 (Sunrise 2210 X Thailand), and non-transgenic Thailand (T), were established in this study. The TC seedlings of transgenic TN-2 with hermaphrodite sex produced uniform fruits of high quality and yield (Figure 4e) and are highly beneficial for practical cultivation, collection, and marketing. The newly developed hybrid has no significant differences from the non-transgenic hybrid Tainung No.2 in fruit size, shape, and sweetness (Appendix A). Three selected hermaphrodite transgenic TN-2 (#1, #5, and #7) were micro-propagated by TC, and their clones were used for further investigation.

### 3.5. Broad-Spectrum Resistance to PRSV Strains under Greenhouse Conditions

To test the resistance effectiveness, TC clones of the selected transgenic line Sunrise 2210 and the new transgenic TN2 #1 (Sunrise 2210 X Thailand) were inoculated with different strains of PRSV from various geographic regions, including Taiwan (YK), Hawaii (HA), Thailand (TH), and Mexico (MX) strains [17,32,33] at the 15 cm stage (with 10 plants for each treatment of transgenic plants of Sunrise 2210, TN-2 #1, and non-transgenic TN-2 control) under greenhouse conditions. All transgenic plants of Sunrise 2210 and TN-2 #1 displayed no symptoms throughout the experimental period (28 days) (Figure 5a), indicating the broad-spectrum resistance of the plants of the transgenic line Sunrise 2210, and the new transgenic TN-2 (Sunrise 2210 X Thailand) to the four PRSV strains. The absence of PRSV in symptomless plants was confirmed by ELISA performed at 28 dpi (Figure 5b) and RT-PCR (Figure 5b). When the clones of transgenic TN-2 #5 and # 7 were tested in the same way, complete resistance was also noticed.

### 3.6. Resistance Evaluation under Field Conditions

The PRSV resistance of transgenic TN-2 # 1 was further evaluated under field conditions. The field tests were performed in certified confined experimental fields for one year, with 60 plants of selected TN2 #1. The results showed that the TC clone of transgenic TN-2 #1 carrying the hemizygous transgene showed no viral infection for one year and produced 60 kg of high-quality fruits as compared to the non-transgenic Tainung No.2 papaya plants (Figure 5e and Appendix A), which were 100% infected with PRSV four months after transplanting and no healthy fruits produced (Figure 6). The absence of virus infection was noticed by symptomless condition and confirmed by the negative results of ELISA. Our results indicated that the hermaphrodite clones of TN-2 #1 provide complete resistance to PRSV in Taiwan under natural conditions.

## 4. Discussion

Papaya fruits can be harvested 8–10 months after planting in the orchards and can continue to produce for 2–3 years. In traditional breeding, the identification of the sex of papaya plants can only be recognized at the flowering stage, which takes about four months after transplanting in the field. The molecular markers in this study were applied in the seedling stage to select individuals with the CP-transgene and hermaphrodite sex to save time and cost. The breeding process contained six times backcrossing and three times selfing to fix the homozygosity of PRSV CP transgene in the maternal Sunrise cultivar of the hybrid Tainung No.2., assisted by various molecular markers. After the final hybridization with another parental cultivar, Thailand, the new commercial cultivar of transgenic TN-2 with hemizygous CP-transgene and broad-spectrum resistance to different PRSV strains was generated. The best hermaphrodite individuals were selected and micropropagated by TC for practical applications. The total breeding process took more than ten years, even with the assistance of molecular markers. The TC clones of the selected hermaphrodite TN-2 provide broad-spectrum resistance to different geographic PRSV strains and produce high-quality, uniform fruits. Our efforts solve the female problem of the original TN-2 transgenic lines and provide a novel cultivation model for papaya plantations in Taiwan and other regions.

The single copy of PRSV CP transgene in the genome of the original transgenic line 16-0-1 was previously validated by Southern blotting and PCR with a marker specific to the CP-transgene [17,19]. In this investigation, the previously reported flanking sequences [19] and improved papaya genome assembly [23] allowed us to map the transgene to an intergenic non-coding region of papaya chromosome 3 in between an unannotated gene (gene ID LOC110819880) and a predicted gene of DEAD-box ATP-dependent RNA helicase 5 (gene ID LOC110819887) (Figure 1b). The newly extended flanking sequences obtained through the genomic alignment also helped to develop new flanking markers, which are more widely separated than the markers previously used [19]. Since the non-transgenic papaya and hemizygous transgene plants contain alleles without the CP-insert, the flanking sequences can easily amplify a genomic product with the previous flanking primers or newly extended flanking primers (Figure 1 and Figure 2). The plants with homozygous CP-transgene cannot generate the product using the flanking markers since both alleles of the diploid chromosome containing T-DNA insert, which is about 5.5 Kb [19], are too long for conventional PCR to amplify a product (Figure 1 and Figure 2). The newly extended flanking markers confirm the results using the primers designed from previously identified short flanking sequences and can precisely verify the zygosity status of the transgene.

Sex determination facilitated by genotyping the hermaphrodite-specific W11 locus [27] was conducted at the seedling stage during each breeding step. The applications of the CP-transgene-specific marker, the flanking sequence markers, and the sex-linked marker precisely screen the progenies with the CP-transgene, hemi- or homozygosity, and hermaphrodite sex at the seedling stage. These significantly shorten the breeding process; one crossing takes at least a year, since papaya is a fruit crop.

PRSV CP-transgene with other nuclear DNA fragments of the transgenic papaya SunUp was inserted into papaya chromosome 5 through particle bombardment [23]. Although the transgenic insertion in the SunUp genome is 1.6 Mb long, containing transgenes, plastid genomic fragments, and mitochondrial genomic fragments, no gene expression change was detected in transcriptomic analysis [23]. Our PRSV CP transgenic line 16-0-1 was generated by *Agrobacterium*-mediated transformation [16], which does not introduce DNA fragments in addition to T-DNA insert. In this study, the insertion of the PRSV YK CP transgene in the non-coding region of chromosome 3 implies that all the horticultural characteristics of line 16-0-1 are unaffected. Moreover, no difference in the horticultural traits associated with the CP transgene was ever observed during our breeding process, evidencing that PRSV YK CP transgene does not influence the horticultural properties of the transgenic TN-2 hybrid derived from Sunrise 2210 X Thailand. A small section of genomic DNA (15-nts) was lost at the CP-transgene insertion site, which is a common phenomenon for the agrobacterium-mediated T-DNA transformation [34,35].

Micropropagation techniques for papaya have been used to generate homogenous papaya clones of the same genotype [17,28,36]. Taiwan is the only country that uses tissue-cultured papaya seedlings for plantation, and currently, around 80% of orchards under netting have changed to this approach. Our selected hermaphrodite transgenic TN-2 #1, 5, and 7 can be propagated in high throughput by the well-established tissue culture technology to scale up its commercial production. Because of the devastating PRSV infection, most of the papaya in Taiwan has been grown under UV-resistant nylon netting to exclude the access of aphid vectors for preventing PRSV infection [4], which is costly and environmentally hazardous after burning or abandonment. Our new transgenic clones can be grown under natural conditions without netting.

Although the PRSV-resistant transgenic line 16-0-1 of TN-2 hybrid has been developed [17], the practical application of line 16-0-1 is limited because its female fruits are unacceptable to consumers and growers. Through marker-assisted breeding of this study, we obtained the hermaphrodite TN-2 carrying hemizygous transgene and thus solved the female problem of transgenic line 16-0-1. In addition, individual clones of the selected hermaphrodite transgenic TN-2 micropropagated by tissue culture produce homogenous top-quality fruits superior to the seedlings from seeds with sex segregation. The broad-spectrum resistance of the hermaphrodite transgenic TN-2 papaya retains the criterion of line 16-0-1 [17] with broad-spectrum resistance to PRSV strains from different geographic origins. Our transgenic TN-2 hybrid is superior to the transgenic hybrid Rainbow of Hawaii, which provides resistance only to Haiwaii isolates of PRSV [11,13,15], and can not be used in other geographic areas. Thus, our TN #1, 5, and 7 clones provide a new model of papaya cultivation in Taiwan and elsewhere.

According to the regulatory requirements of genetically modified crops in the European Union and several Asian countries, tracking information on the transgenic line is necessary. The transgenic TN-2 papaya resistant to PRSV in this study derived from the lineage with patented transgene-specific or event-specific detections (US8258282B2), food safety evaluation [37], and a biosafety certification approval by China (2018 Certificate No.174), is ready to boost the global papaya industry.

## Figures and Tables

**Figure 1 viruses-16-00823-f001:**
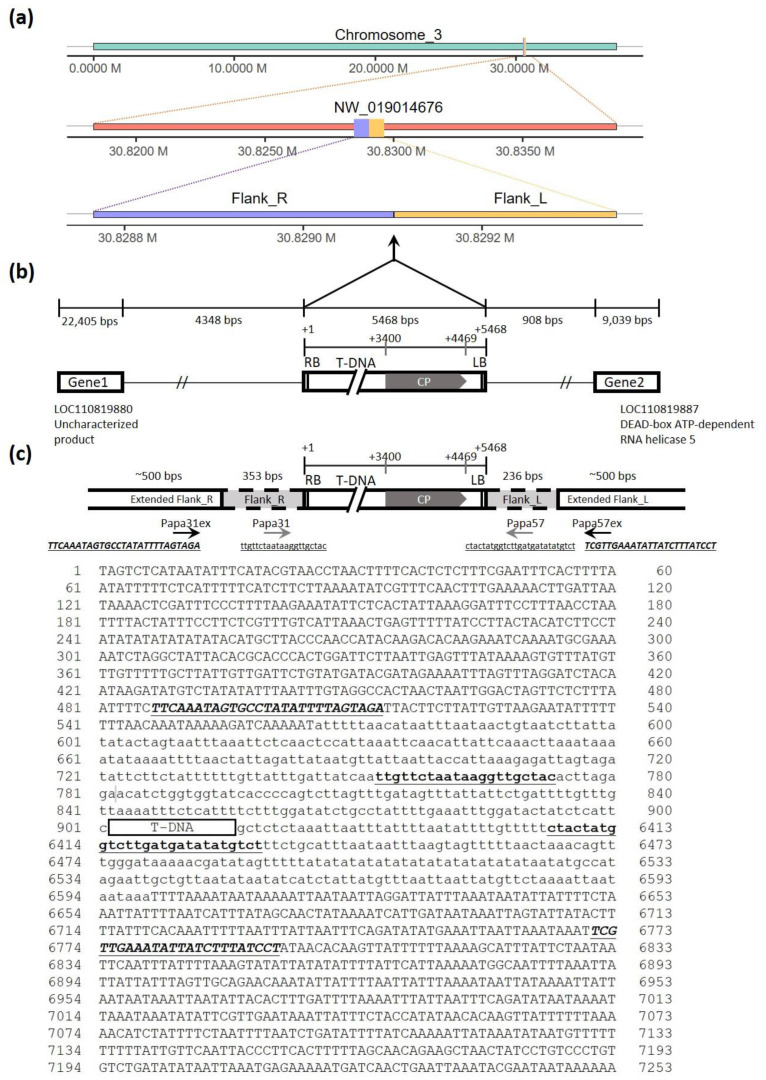
Mapping of PRSV CP transgene in the papaya genome and generation of new molecular markers based on extended flanking sequences. (**a**) The diagram presents CP-transgene insertion in chromosome 3 of the papaya genome (**top**), in contig NW_019014676 (**middle**), using the right (Flank_R) and left (Flank_L) flanking sequences identified by Fan et al. 2009 (**bottom**) for alignment. (**b**) The genetic map indicates the transgene insert in the intergenic region between LOC110819880 (an uncharacterized gene) and LOC110819887 (DEAP-box ATP-dependent RNA helicase). (**c**) The sequences of primers Papa31 and Papa57 designed from flanking sequences [19] (in lower case) and Papa31ex and Papa57ex designed from the extended flanking sequences (in upper case) from this study for zygosity assessment are shown. The positions of primers for zygosity assessment are underlined.

**Figure 2 viruses-16-00823-f002:**
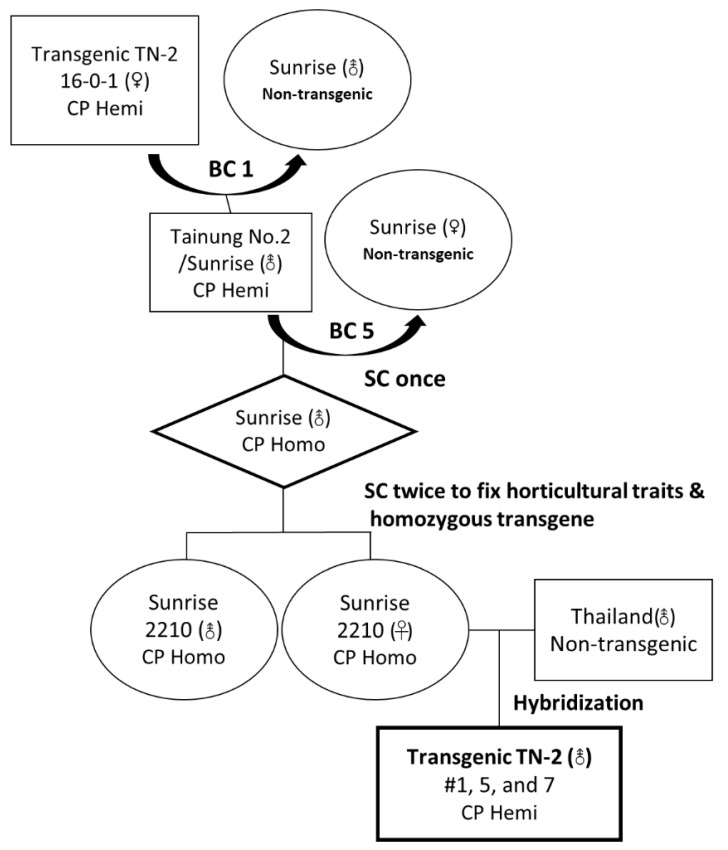
Generation of Sunrise papaya with homozygous transgene and breeding of hermaphrodite papaya hybrid Tainung No.2 (TN-2) carrying the hemizygous PRSV CP transgene. The hemizygous line 16-0-1 of PRSV-resistant TN-2 female (♀) papaya plant carrying PRSV YK CP transgene (CP Hemi) was selected as the source for breeding into a transgenic hermaphrodite (⚨) TN-2 hybrid. The hermaphrodites with hemizygous transgene were generated by backcrossing 16-0-1 TN-2 with non-transgenic parental Sunrise. By a total of six times of backcrossing (BC) and three times of self-crossing (SC), the homozygous transgene (CP Homo) was fixed in the Sunrise cultivar. The selected females of maternal Sunrise 2210 carrying homozygous transgene were propagated by tissue culture and were used to cross with the hermaphrodite Thailand (T) cultivar to generate the hybrid TN-2 papaya with hemizygous (Hemi) transgene. Three hermaphroditic individuals of the transgenic TN-2 #1, 5, and 7 were selected and micro-propagated by tissue culture for further evaluation.

**Figure 3 viruses-16-00823-f003:**
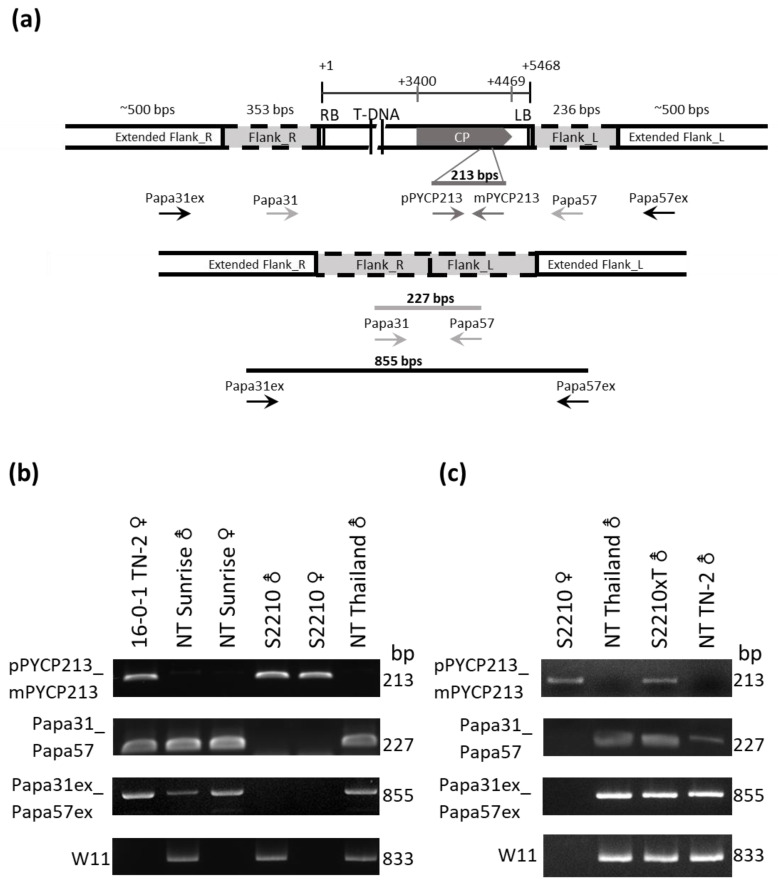
Molecular-marker assisted breeding for generating homozygous transgenic Sunrise and hemizygous transgenic hybrid Tainung No. 2 (TN-2) carrying PRSV CP. (**a**) The PCR primer pairs designed from the CP transgene (pCYCP213/mPYCP213), and flanking sequences (Papa31/Papa57 and Papa31ex/Papa57ex) are shown in the allelic diagram of a hemizygote plant. (**b**) Positive PCR products of 213 bp with the CP primer pair and 227 bp and 855 bp from the two flanking sequence primer pairs revealed the hemizygous transgene in the transgenic line 16-0-1 of TN-2. Negative CP product and positive amplified products of 227 bp and 855 bp by the two flanking sequence primer pairs reflected non-transgenic (NT) status. Positive CP product and negative amplifications of the two flanking sequence primer pairs denoted the homozygous transgene in the transgenic line Sunrise 2210. A positive product of 833 bp with sex marker W11 indicated hermaphrodite sex. (**c**) Genotyping results of hemizygous TN-2 hybrid from homozygous Sunrise 2210 crossed with non-transgenic Thailand (S2210xT) are shown, and non-transgenic Thailand and TN-2 were used as negative controls.

**Figure 4 viruses-16-00823-f004:**
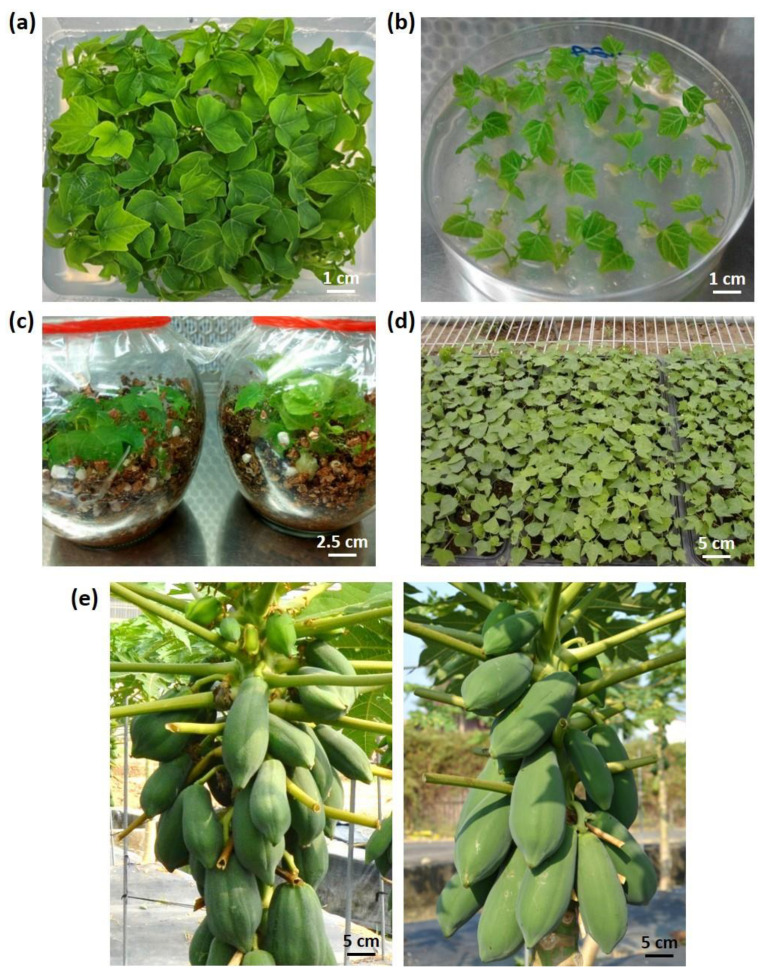
Tissue culture propagation of hermaphroditic individuals of transgenic Tainung No.2 papaya. (**a**) Multiple shoots propagation. (**b**) Induction of rooting. (**c**) Root development in the rooting medium. (**d**) High-throughput production of papaya clones of hermaphroditic PRSV CP transgenic TN-2 #1. (**e**) Comparison between hermaphroditic fruits of non-transgenic (**left**) and transgenic Tainung No.2 #1 papaya (**right**) seven months after planting under field conditions.

**Figure 5 viruses-16-00823-f005:**
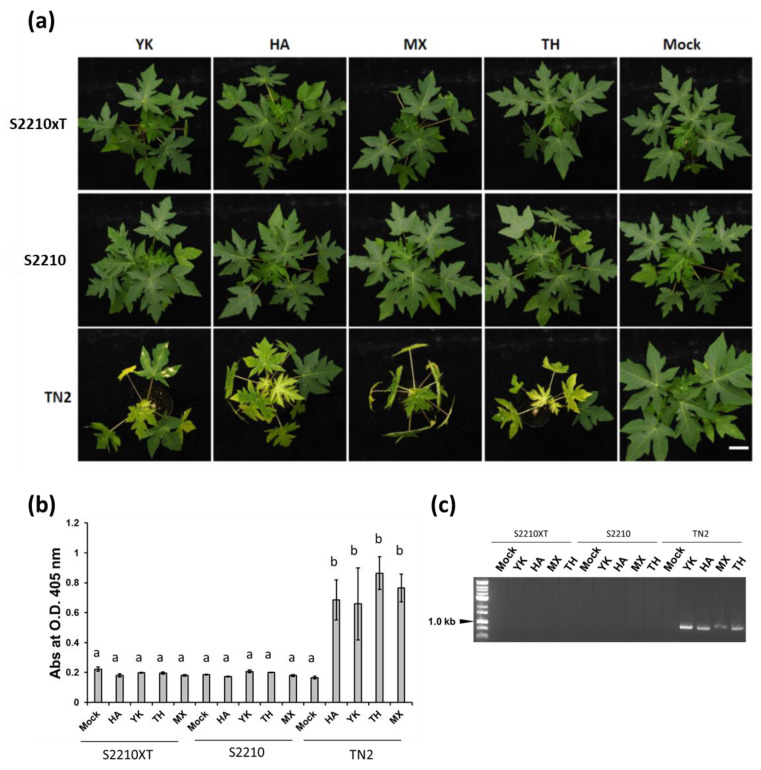
Resistance evaluation of the selected hermaphrodite transgenic hybrid TN-2 against different geographic PRSV strains under greenhouse conditions. (**a**) Different PRSV strains (HA: Hawaii, MX: Mexico, TH: Thailand, YK: Taiwan, and Mock: buffer) were used to inoculate papaya plants of transgenic Sunrise 2210 (S2210), Transgenic TN-2 (Sunrise 2210 X Thailand, S2210xT), and non-transgenic papaya (TN2) control (*n* = 10, scale bar: 5 cm). (**b**) Three randomly picked symptomless plants 28 days after inoculation were detected by ELISA using PRSV CP antiserum with reads recorded 30 min after substrate addition. The one-way analysis of variance (ANOVA) was used for each dataset (each bar) and means (*n* = 3) in each column followed by the same letters are not significantly different (*p*-value > 0.05, Holm Sidak test). (**c**) RT-PCR detection using PRSV CP specific primers.

**Figure 6 viruses-16-00823-f006:**
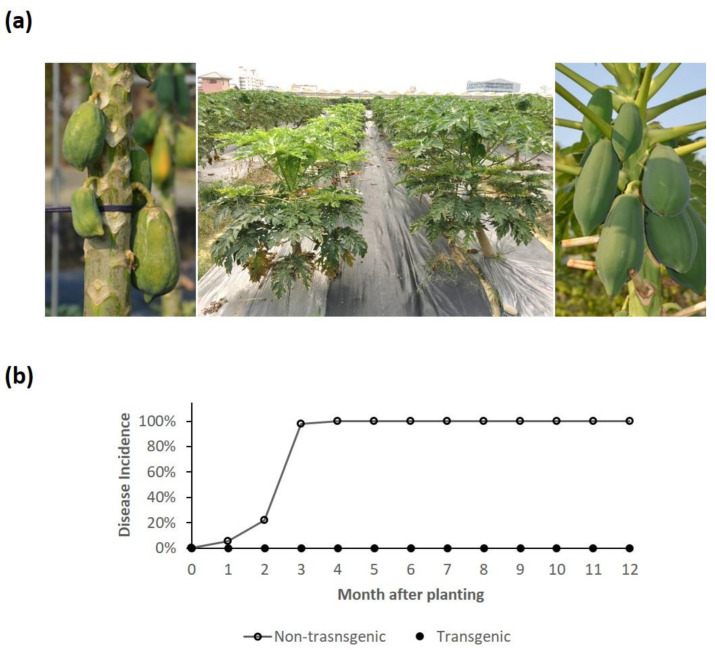
Virus-resistance evaluation of selected tissue-cultured clones of hermaphrodite transgenic papaya hybrid TN-2 # 1 under field conditions. (**a**) Hermaphrodite transgenic hybrid clones of TN-2 # 1 (right row in the middle) versus non-transgenic TN-2 (left row in the middle) in the field at four months after planting. A close-up view of the individual plant was recorded for a non-transgenic TN-2 papaya (**left**) and a representative hermaphrodite transgenic TN-2 plant (**right**) at seven months after planting. (**b**) The disease incidences were recorded during the one-year test period (November 2020–October 2021), as checked by symptom development and ELISA confirmation.

## Data Availability

The original contributions presented in the study are included in the article/Appendix A, further inquiries can be directed to the corresponding author/s.

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
