# Peer review of "Mapping the CP-Transgene Insert in the Papaya Genome and Developing a Hermaphrodite Transgenic Hybrid with Broad-Spectrum Resistance to Papaya Ringspot Virus"

_viruses, 2024, doi:10.3390/v16060823_

Round 1

Reviewer 1 Report

Comments and Suggestions for Authors

This study presents a novel plantation model of transgenic papaya with broad-spectrum resistance to Papaya ringspot virus (PRSV) strains to produce high-quality papaya fruits. Authors designed a molecular mapping approach for identifying genomic location of CP transgene insert in papaya genome and utilized molecular markers assisted breeding for generating homozygous transgenic Sunrise and hemizygous transgenic hybrid Tainung No. 2 (TN-2) carrying PRSV CP. Furthermore, micropropagation and resistance to PRSV strains for selected TN-2 hermaphrodite individuals is demonstrated for practical application. Overall, this manuscript is novel, clearly written, and well conceptualized, and I would recommend it suitable for the journal. There are few minor comments which should be addressed to facilitate clear understanding.

1-    Fig-5b, please explain what the alphabetical symbols a & b are indicating for ELISA data.

2-    Fig-5c, there should be endogenous control (beta actin) for RT-PCR data.

3-    Fig-6b, please explain in detail how disease incidence is calculated. Also, please include the rationale for including time window of Nov 2020 to Oct 2021.

4- Please include the details of template concentrations used for PCR wherever applicable in methods.

5-  I would suggest including a table with details of all primers used in this study along with expected amplicon sizes wherever applicable.

Author Response

We deeply appreciate the time and effort that each of the reviewers has dedicated to enhancing our manuscript. Their affirmative comments and insightful suggestions have been thoroughly considered and integrated into the revised version. We present our responses below:

This study presents a novel plantation model of transgenic papaya with broad-spectrum resistance to Papaya ringspot virus (PRSV) strains to produce high-quality papaya fruits. The authors designed a molecular mapping approach for identifying the genomic location of CP transgene insert in the papaya genome and utilized molecular markers assisted breeding for generating homozygous transgenic Sunrise and hemizygous transgenic hybrid Tainung No. 2 (TN-2) carrying PRSV CP. Furthermore, micropropagation and resistance to PRSV strains for selected TN-2 hermaphrodite individuals are demonstrated for practical application. Overall, this manuscript is novel, clearly written, and well conceptualized, and I would recommend it suitable for the journal. There are few minor comments which should be addressed to facilitate clear understanding.

Response: We thank the reviewer for the affirmative comments.

1-    Fig-5b, please explain what the alphabetical symbols a & b are indicating for ELISA data.

Response: Fig-5b, the meaning of alphabetical symbols a & b are described at lines 684-686 in the figure legend. “Each column followed by the same letters are not significantly different (p-value < 0.05, Holm Sidak test).” Consequently, the columns with “a” above indicate that the datasets are not significantly different. There is no revision for this query.

2-    Fig-5c, there should be endogenous control (beta actin) for RT-PCR data.

Response: Fig-5c, the endogenous control for RT-PCR is critical to validate the procedure. However, we determined whether the papaya is infected by PRSV or not mainly based on the symptoms and ELISA. The RT-PCR simply served as additional evidence to consolidate the absence of the virus. We do not use endogenous control for virus detection because the viral RNA is more abundant than any endogenous gene. With our negative and positive controls, the RT-PCR results further confirm the absence of the challenge virus.  There is no revision for this comment.

3-    Fig-6b, please explain in detail how disease incidence is calculated. Also, please include the rationale for including time window of Nov 2020 to Oct 2021.

Response: Fig-6b shows the calculation method of disease incidence based on the development of the symptoms according to Bau et al., 2004 (Ref 18), indicating lines 267-268. The reason for conducting the field test from November 2020 to October 2021 is that the papaya needs to be cultivated for about 7 months to harvest the fruit after planting. We designed the field experiment for a year period to observe the complete growth phase of papayas. November is the best time to transplant papaya seedlings in the field under Taiwan’s conditions, which can avoid damage by heavy rains or tropical storms (lines 260-263).

4- Please include the details of template concentrations used for PCR wherever applicable in methods.

Response: In the breeding process PCR, the final concentration of papaya genomic DNA was 0.5~1.0 ng/μl (lines 225-227). For RT-PCR validation, around 1 μg total RNA was subjected to reverse transcription for each sample, and then cDNA was 10-fold diluted for PCR (line 253).

5-  I would suggest including a table with details of all primers used in this study along with expected amplicon sizes wherever applicable.

Response: Thank you for your suggestion. The major primers designed in this study are clearly described in Figure 1, lines 195-196. The sources of other primers, such as CP primers (Ref 30), flanking sequence primers (Ref 19), and sex-linked primers (Ref 27), are clearly cited with relevant references since they were not initially designed in this study. Thus, a table list for all primers is not necessary. Lines 197, 201, and 204 originally describe individual amplicon sizes. There are no revisions for this query.

Reviewer 2 Report

Comments and Suggestions for Authors

This paper reports on the generation of transgenic papaya plants highly resistant to a wide range of papaya ringspot virus (PRSV) strains. This is an example of excellent biotechnology work aimed at providing a highly demanded papaya variety to the market. Using a previously obtained transgenic line, 16-0-1, the authors used classical breeding assisted by the use of molecular markers to obtain a hermaphrodite transgenic line, TN-2, that fully met the desired traits, including strong resistance to PRSV. The work is described in full detail, the discussion is relevant to the results presented. The paper is well written, the figures adequately reflect the content of the paper. In summary, I do not see any serious drawbacks in this manuscript. In my opinion, the paper can be published in its present form.

Author Response

We deeply appreciate the time and effort that each of the reviewers has dedicated to enhancing our manuscript. Their affirmative comments and insightful suggestions have been thoroughly considered and integrated into the revised version.

This paper reports on the generation of transgenic papaya plants highly resistant to a wide range of papaya ringspot virus (PRSV) strains. This is an example of excellent biotechnology work aimed at providing a highly demanded papaya variety to the market. Using a previously obtained transgenic line, 16-0-1, the authors used classical breeding assisted by the use of molecular markers to obtain a hermaphrodite transgenic line, TN-2, that fully met the desired traits, including strong resistance to PRSV. The work is described in full detail, the discussion is relevant to the results presented. The paper is well written, the figures adequately reflect the content of the paper. In summary, I do not see any serious drawbacks in this manuscript. In my opinion, the paper can be published in its present form.

Response: We thank the reviewer for the affirmative comments.